# Finite-Time Analysis of Stochastic Nonconvex Nonsmooth Optimization on the Riemannian Manifolds

**Emre Sahinoglu**
Northeastern University
sahinoglu.m@northeastern.edu

**Youbang Sun**
Tsinghua University
ybsun@mail.tsinghua.edu.cn

**Shahin Shahrampour**
Northeastern University
s.shahrampour@northeastern.edu

## Abstract

This work addresses the finite-time analysis of nonsmooth nonconvex stochastic optimization under Riemannian manifold constraints. We adapt the notion of Goldstein stationarity to the Riemannian setting as a performance metric for nonsmooth optimization on manifolds. We then propose a Riemannian Online to NonConvex (RO2NC) algorithm, for which we establish the sample complexity of $O(\epsilon^{-3}\delta^{-1})$ in finding $(\delta, \epsilon)$-stationary points. This result is the first-ever finite-time guarantee for fully nonsmooth, nonconvex optimization on manifolds and matches the optimal complexity in the Euclidean setting. When gradient information is unavailable, we develop a zeroth order version of RO2NC algorithm (ZO-RO2NC), for which we establish the same sample complexity. The numerical results support the theory and demonstrate the practical effectiveness of the algorithms.

## 1 Introduction

Gradient-based iterative optimization algorithms are frequently used as numerical solvers for machine learning problems, many of which deal with search spaces with manifold structures. This includes deep learning [61], natural language processing [34], principal component analysis (PCA) [19], dictionary learning [11, 56], Gaussian mixture models [29] and low-rank matrix completion [27]. Riemannian optimization methods [1, 5, 54] offer a toolkit to solve optimization problems with manifold constraints and have attracted great interest due to their wide range of applications. However, unfortunately, conventional Riemannian optimization algorithms require access to the derivative of a *smooth* function, often falling short in highly nonsmooth, nonconvex problems such as training neural networks.

In this paper, we consider stochastic Riemannian optimization of an objective function $f : \mathcal{M} \to \mathbb{R}$,

$$\min_{x \in \mathcal{M}} \left\{ f(x) := \mathbb{E}_\nu[F(x, \nu)] \right\}, \tag{1}$$

where $F$ is a stochastic *nonsmooth* objective function, defined on a $d$-dimensional complete manifold $\mathcal{M}$ that can be embedded in the Euclidean space, and $\nu$ corresponds to random data samples. Many important problems can be formulated as nonsmooth Riemannian optimization over smooth manifolds, including sparse PCA, compressed modes in physics, unsupervised feature selection, and robust low-rank matrix completion [10, 30]. This problem setting is also commonly encountered in deep neural networks, where optimization algorithms must be able to cope with deep nonlinear layers with ReLU activations [2, 20, 21, 55, 69].

39th Conference on Neural Information Processing Systems (NeurIPS 2025).

Table 1: Convergence rates of various algorithms over nonsmooth, nonconvex objectives: Composite objectives are in the form of $f(x) + h(x)$, where $f$ is smooth and $h$ is possibly nonsmooth and convex. Fully nonsmooth objectives can be written as $f(x) = E_\nu[F(x, \nu)]$, where $F$ is nonsmooth and nonconvex. Our work is the first to provide the finite-time analysis of the *fully* nonsmooth setting. [†] and [‡] indicate that the objective is defined on Stiefel manifold or a compact manifold, respectively.

| REFERENCE | METHOD | SETTING | OBJECTIVE | CONVERGENCE |
|---|---|---|---|---|
| [23] | SUBGRADIENT | DETERMINISTIC | NONSMOOTH | ASYMPTOTIC |
| [10] | PROXIMAL GRADIENT | DETERMINISTIC | COMPOSITE [†] | $O(\epsilon^{-2})$ |
| [59] | PROX. GRAD. - SPIDER | STOCHASTIC | COMPOSITE [†] | $O(\epsilon^{-3})$ |
| [44] | R-ADMM | DETERMINISTIC | COMPOSITE [‡] | $O(\epsilon^{-4})$ |
| [16] | AUG. LAGRANGIAN | DETERMINISTIC | COMPOSITE [‡] | $O(\epsilon^{-3})$ |
| [16] | AUG. LAGRANGIAN | STOCHASTIC | COMPOSITE [‡] | $\tilde{O}(\epsilon^{-3.5})$ |
| [50] | SMOOTHING | DETERMINISTIC | COMPOSITE [‡] | $O(\epsilon^{-3})$ |
| [50] | SMOOTHING | STOCHASTIC | COMPOSITE [‡] | $O(\epsilon^{-5})$ |
| **OUR WORK** | SUBGRADIENT | STOCHASTIC | NONSMOOTH | $O(\delta^{-1}\epsilon^{-3})$ |

There exists a scant literature on the theory of nonsmooth Riemannian optimization, mainly focusing on the asymptotic results and leaving the *finite-time analysis* unexplored. In fact, even in the Euclidean setup, finite-time results for the nonsmooth, nonconvex setting have been studied only recently in the last few years. This is perhaps not surprising; the finite-time analysis of smooth, nonconvex optimization is usually carried out by finding a stationary point $x$ such that $\|\nabla f(x)\| \leq \epsilon$. However, for nonsmooth, nonconvex functions, finding such an $\epsilon$-stationary point is in fact intractable. Instead, the notion of Goldstein stationarity has recently been analyzed by [66] to provide non-asymptotic results. Goldstein stationarity does not directly evaluate $\|\nabla f(x)\|$ and instead considers the convex hull of subgradients of points in the $\delta$-neighborhood of $x$.

To address nonsmooth problems on *Riemannian* manifolds, this stationarity criterion can be adapted to the Riemannian setting using the Riemannian $\delta$-subdifferential definition of [32]. However, the finite-time analysis of finding such $(\delta, \epsilon)$-stationary points on Riemannian manifolds is a tantalizing challenge that has remained elusive.

**Our Contributions.** In this paper, we address the *finite-time* analysis of *nonsmooth, nonconvex stochastic Riemannian* optimization:

- We adapt the notion of Goldstein stationarity [45, 66] to the Riemannian setting, providing a metric to evaluate the finite-time performance of nonsmooth, nonconvex optimization on manifolds. We then propose a Riemannian Online to NonConvex (RO2NC) algorithm that can be analyzed using regret bounds in online optimization. Our proposed algorithm can utilize retraction as a computationally efficient alternative for exponential mapping.

- We theoretically prove that RO2NC achieves the sample complexity of $O(\delta^{-1}\epsilon^{-3})$ in finding $(\delta, \epsilon)$-stationary points. We first establish this result using parallel transports (Theorem 3.1) and then extend it to projections (Theorem 3.3). This rate is the *first ever finite-time result for the fully nonsmooth, nonconvex optimization on manifolds* and matches the optimal sample complexity in the Euclidean setting [13]. It is derived under mild technical assumptions applicable to standard manifolds (e.g., Stiefel, Hadamard, Grassmann) without the need for restrictive assumptions on the objective function (e.g., weak convexity), which is assumed to be Lipschitz continuous.

- Extending our results to the *zeroth order* oracle setting, we show that ZO-RO2NC can achieve the same rate (Theorem 4.2) using a gradient estimator relying solely on stochastic function value queries. Our proposed gradient estimator samples vectors on the tangent space instead of the manifold, resulting in more computational efficiency without costly manifold-specific computations. We show that the gradient estimator satisfies certain distance bounds to the Goldstein subdifferential set, which is sufficient to achieve the Goldstein stationarity criterion. The main merit of this result is streamlining the zeroth order analysis without recourse to more challenging alternative estimators that require the calculation of volume and surface of manifolds, which could be computationally demanding.

## 1.1 Highlights of the Technical Analysis

**RO2NC Algorithm Design.** Developed by [13], O2NC is an optimal algorithm for finding Goldstein stationary points of nonsmooth, nonconvex objectives in the Euclidean space. O2NC works based on the interplay of two algorithms. At each epoch, an action $\Delta_t$ is generated via an online learning algorithm and is used to update the variable $x_{t+1} = x_t + \Delta_t$. Then, a gradient $g_t$ is evaluated at a random point $w_t = x_t + s\Delta_t$ with $s \sim \text{unif}[0,1]$ and is given to the online learning algorithm as a feedback to update $\Delta_t$. (i) One difficulty in the Riemannian setting is that $x_{t+1} = x_t + \Delta_t$ does not ensure feasibility. We must keep the iterates $\{x_t\}$ on the manifold $\mathcal{M}$, which requires the use of retractions in the updates, i.e., $x_{t+1} = \text{Retr}_{x_t}(\Delta_t)$. (ii) Also, since the action and feedback of the online learning algorithm belong to different tangent spaces, we must transport these vectors via parallel transports and/or projections. (i) and (ii) together introduce a technical challenge different from the Euclidean setting: we can no longer rely on the classical regret analysis in online optimization to streamline the intra-epoch analysis. Instead, we must creatively choose the right benchmark action at each epoch using projections and/or parallel transports to judiciously analyze the error terms caused by the manifold geometry.

**Adapting Goldstein Stationarity to Riemannian Setting.** In the Euclidean setting, the Goldstein subdifferential set is defined as a convex hull of Euclidean differential sets at different points. In the Riemannian setting, subdifferential sets at different points possibly belong to different tangent spaces, so it is required to transport those sets to a common tangent space. We choose the parallel transport operation in the definition of Riemannian Goldstein subdifferential set as it preserves the length of the vectors unlike the projection operation. Upper bounding the minimum norm element in the Goldstein differential set requires transportation of gradients at different points, where the distortion caused by parallel transport along the closed loops is analyzed with the curvature of the manifold (c.f. Lemma 2.9). Using a similar approach, we derive an upper bound on the distance between zeroth order gradient estimates and Goldstein subdifferential set by quantifying the effect of parallel transport along geodesic triangles (c.f. Lemma 4.1).

**Zeroth Order Riemannian Gradient Estimator.** In the Euclidean setting, the analysis of zeroth order gradient estimator for nonsmooth objectives is carried out through smoothing. The key is that the gradient of the smoothed objective $f_\delta$ is estimated by a stochastic gradient estimator $g_\delta$, such that $\mathbb{E}[g_\delta(x)] = \nabla f_\delta(x)$ [45, Lemma E.1]. Extending this approach to the Riemannian setting introduces a great challenge. The smoothed objective $f_\delta$ necessitates the volume calculation of a manifold set, which is computationally expensive. To make the estimation practical, we define an objective $h_\delta$ based on efficiently sampling a vector in the tangent space, such that $\mathbb{E}[g_\delta(x)] = \text{grad}h_\delta(x)$, where $g_\delta$ is the Riemannian gradient estimator. However, the inclusion property $\mathbb{E}[g_\delta(x)] \in \partial_\delta f(x)$ of the Euclidean setting does not hold anymore in the Riemannian setting. We address this technical challenge by deriving an upper bound on the distance between these two terms as a function of the curvature and $\delta$ (c.f. Lemma A.5). We further compute a bound on the Lipschitz constant of $h_\delta$ (Lemma A.3) and show that the $O(\delta^{-1}\epsilon^{-3})$ convergence rate is preserved.

## 1.2 Literature Review

**Nonsmooth, Nonconvex Techniques in the Euclidean Setting.** Given that calculating an $\epsilon$-stationary point for nonsmooth, nonconvex functions is intractable in general, various assumptions or conditions have been proposed in the literature (e.g., weak convexity [8, 14]). In recent years, the finite-time convergence analysis of the Goldstein stationarity, proposed by [66], has gained much interest. Their algorithm is guaranteed to reach a $(\delta, \epsilon)$-stationary point with the complexity of $O(\epsilon^{-4}\delta^{-1})$. [13] improved this rate to $O(\epsilon^{-3}\delta^{-1})$ with the introduction of an online to nonconvex conversion method. The method was then extended to various settings, such as decentralized optimization [51].

For some optimization applications, gradient evaluations are not possible, and only zeroth order information is available. To solve these problems, [40] proposed a gradient estimator, which is calculated at point $x$ and requires evaluation of $f(x + tv)$. In nonsmooth, nonconvex optimization, the convergence rate for finding $(\delta, \epsilon)$-stationary points with gradient-free methods was analyzed by [45]. Later [37] proved that the optimal dimensional dependence of $O(d)$ in the zeroth order setting could be achieved with O2NC.

**Manifold Optimization.** Due to the unique properties of manifolds, many Euclidean optimization algorithms have been adapted to the Riemannian setting. Utilizing manifold operations, these

methods are able to match the convergence rates of their Euclidean counterparts. These Riemannian algorithms include gradient descent methods [3, 65], projection-free methods [63, 64], and accelerated methods [36, 48]. The convergence results also cover various settings, such as min-max [35, 49, 68], variance reduction [54], online optimization [6, 46, 52, 53, 62], and decentralized optimization [7, 57]. However, the majority of the studies here focus on *smooth* and sometimes geodesically convex objective functions, leaving the nonsmooth optimization relatively underexplored.

For the gradient-free optimization setting, zeroth order methods have been adapted from Euclidean to the Riemannian setting [42, 43, 60]. The Riemannian zeroth order methods have also been extended to different settings, such as online Riemannian algorithms [46, 52, 62] and accelerated algorithms [26].

**Nonsmooth, Nonconvex Optimization on Manifolds**. The optimization task becomes more challenging when the objective function is nonsmooth and nonconvex over the Riemannian manifold. Some techniques have been developed and analyzed in nonsmooth, nonconvex optimization, such as (i) subgradient-oriented methods, (ii) proximal point methods, and (iii) operator splitting methods. In subgradient-oriented methods [4, 17, 18, 23, 25, 32], at iteration $t$ a direction $g_t$ in the tangent space of the manifold is calculated from the current or previous subgradient evaluations. The next iterate $x_{t+1}$ is then calculated based on manifold operations, such as retractions or exponential mappings. For proximal point algorithms [9, 15, 28, 33], the algorithm solves a sub-problem in each iteration. The objectives for the sub-problems are formulated by an approximation of the original function combined with an additional penalty term. However, in some cases, solving the sub-problems is just as difficult as the original problem, making these methods difficult to implement. The operator-splitting methods [39] divide the original problem into several sub-problems that are easy to solve using techniques such as the alternating direction method of multipliers (ADMM). These methods either lack convergence guarantees [38] or need further technical conditions [67].

## 2 Preliminaries

In this section, we first provide a brief introduction on manifolds and useful notations/definitions. Then, we introduce Goldstein stationarity and state our technical assumptions.

### 2.1 Background on Manifold Optimization

We consider the optimization task in equation 1 defined on $\mathcal{M}$, which is an embedded submanifold in the Euclidean space. We denote by $T_x\mathcal{M}$ the tangent space of the manifold at point $x$. We denote the sphere (and ball) with radius $r$ on $T_x\mathcal{M}$ by $\mathbb{S}_{T_x\mathcal{M}}(r)$ (and $\mathbb{B}_{T_x\mathcal{M}}(r)$), respectively. The set of pairs $(x, \xi_x)$ where $\xi_x \in T_x\mathcal{M}$ is referred to as the tangent bundle, denoted by $T\mathcal{M}$. Similarly, the set of pairs $(x, s_x)$ such that $\langle s_x, \xi_x \rangle = 0$ for every $\xi_x \in T_x\mathcal{M}$ is called the normal bundle, denoted by $N\mathcal{M}$. Let $\mathfrak{X}(\mathcal{M})$ denote space of vector fields on $\mathcal{M}$ and $\nabla : T\mathcal{M} \times T\mathcal{M} \to T\mathcal{M}, (\xi, \eta) \to \nabla_\xi \eta \in T\mathcal{M}$ denote the Levi-Civita connection on manifold $\mathcal{M}$. We adapt the Euclidean metric to the embedded manifold and use $\langle \cdot, \cdot \rangle$ and $\| \cdot \|$ to denote the inner product and norm, respectively.

*Geodesics* on the manifolds are generalizations of lines in Euclidean space, curves with constant speed that are locally distance-minimizing. Consequently, we can define the distance between two points on the manifold as the length of geodesic $\gamma$, $\mathrm{dist}(x, y) := \inf_\gamma \int_0^t \|\gamma'(t)\| dt$, where $\gamma(0) = x$ and $\gamma(1) = y$. With the help of geodesics, we next introduce the *exponential mapping* and *retraction* operations. From an optimization perspective, both exponential mapping and retraction are used to traverse manifold $\mathcal{M}$. The exponential mapping on a Riemannian manifold defines a geodesic on the manifold $\gamma(t) = \mathrm{Exp}_x(tv)$ and the distance between $x$ and $\mathrm{Exp}_x(v)$ is $\mathrm{dist}(x, \mathrm{Exp}_x(v)) = \|v\|$. Exponential mappings are hard to compute in general; as mitigation, retractions $\mathrm{Retr}_x(\cdot) : T_x\mathcal{M} \to \mathcal{M}$ are introduced as first-order approximations of exponential mappings and are easier to compute. Based on the definition of distance, we denote by $B(x, \delta) := \{y \in \mathcal{M} : \mathrm{dist}(x, y) \leq \delta\}$ a ball centered at $x$ with radius $\delta$ [32].

Given that the tangent space $T_x\mathcal{M}$ is defined with respect to point $x$, vectors defined in different tangent spaces are not directly comparable. To this end, we can use the notion of *parallel transport*, which is a linear, isometric mapping from one tangent space to another, defining a way to transport the local geometry along a curve. We denote parallel transport along the minimizing geodesic by $P_{x,y}^g : T_x\mathcal{M} \to T_y\mathcal{M}$, which preserves the inner products such that $\langle u, v \rangle = \langle P_{x,y}^g(u), P_{x,y}^g(v) \rangle$

for $u, v \in T_x\mathcal{M}$. As a computationally efficient alternative to parallel transports, we can use vector transports [1], a specific case of which is projection. For $x, y \in \mathcal{M}$ the orthogonal projection onto tangent space can be defined as $\mathrm{Proj}_{T_y\mathcal{M}} : T_x\mathcal{M} \to T_y\mathcal{M}$, mapping $\xi \in T_x\mathcal{M}$ to $\mathrm{Proj}_{T_y\mathcal{M}}(\xi) \in T_y\mathcal{M}$ such that $\langle \mathrm{Proj}_{T_y\mathcal{M}}(\xi), \xi - \mathrm{Proj}_{T_y\mathcal{M}}(\xi) \rangle = 0$. While more efficient than parallel transports, vector transports (and projections) are not necessarily isometric, requiring more delicate analysis.

## 2.2 Goldstein Stationarity on Riemannian Manifolds

Using concepts defined in Riemannian geometry, we next provide the following notions for optimization on manifolds. As a result of Rademacher's theorem and local equivalence of the Riemannian distance with Euclidean distance in a chart, every Lipschitz function defined on a Riemannian manifold $\mathcal{M}$ is differentiable almost everywhere with respect to the Lebesgue measure on $\mathcal{M}$ [32].

**Definition 2.1.** We define the Riemannian subdifferential of $f$ at $x$, denoted by $\partial f(x) := \mathrm{conv}\{\lim_{l \to \infty} \mathrm{grad} f(x_l) : x_l \to x, x_l \in \Omega_f\} \subset T_x\mathcal{M}$, where $\Omega_f : \{x \in \mathcal{M} : f$ is differentiable at $x\}$ and $\mathrm{conv}$ denotes the convex hull operator.

Here, $\mathrm{grad} f(x)$ denotes the Riemannian gradient, defined as the unique tangent vector that satisfies $df(x)[\xi] = \langle \mathrm{grad} f(x), \xi \rangle$ for all $\xi \in T_x\mathcal{M}$. Although the sequence $\{\mathrm{grad} f(x_l)\}_l$ lies in different tangent spaces $\{T_{x_l}\mathcal{M}\}_l$, the limit $\lim_{l \to \infty} \mathrm{grad} f(x_l)$ can still be defined in a weak sense. Specifically, it is characterized as the vector satisfying $\lim_{l \to \infty} \langle \mathrm{grad} f(x_l), X(x_l) \rangle \to \langle \mathrm{grad} f(x), X(x) \rangle$ for any smooth vector field $X$ on $\mathcal{M}$ [23]. For a smooth function $f$ defined on an embedded submanifold in the Euclidean space $\mathrm{grad} f(x) = \mathrm{Proj}_{T_x\mathcal{M}}(\nabla f(x))$. This also holds for the subdifferential sets and the Riemannian subdifferential set is the projection of the Euclidean subdifferential set onto the tangent space at $x$.

Definition 2.1 extends the notion of the Clarke subdifferential to the Riemannian setting, following prior work such as [23, 31]. Among the various subdifferential concepts, the Clarke subdifferential is particularly useful due to its inclusivity and well-behaved analytical properties, which facilitate the study of convergence in nonsmooth optimization. We next provide the definition for $\delta$-subdifferential in the neighborhood of point $x$.

**Definition 2.2.** Let $f$ be a Lipschitz continuous function on $\mathcal{M}$. $\delta$-subdifferential of $x \in \mathcal{M}$ is defined as $\partial_\delta f(x) := \mathrm{cl} \, \mathrm{conv}\{P_{y,x}^g(\partial f(y)) : y \in \mathrm{cl} \, B(x, \delta)\}$, where $\mathrm{cl}$ denotes the closure.

Different from the Euclidean $\delta$-subdifferential set, the Riemannian $\delta$-subdifferential requires a transport operation $P_{y,x}^g$ since the tangent spaces are not identical. Next, we introduce the Riemannian version of Goldstein stationarity.

**Definition 2.3.** Given a Lipschitz continuous function $f : \mathcal{M} \to \mathbb{R}$, a point $x \in \mathcal{M}$ and $\delta > 0$, denote $\big\|\mathrm{grad} f(x)\big\|_\delta := \min\{\|g\| : g \in \partial_\delta f(x)\}$. A point $x$ is called a $(\delta, \epsilon)$-stationary point of $f(\cdot)$ if $\big\|\mathrm{grad} f(x)\big\|_\delta \leq \epsilon$.

This definition generalizes the Euclidean Goldstein stationarity to the Riemannian setting. The main difference is that parallel transport operations and Riemannian subdifferentials are used in $\delta$-subdifferential set due to Riemannian geometry.

Let $X$ and $Y$ be vector fields on $\mathcal{M}$. Since $\mathcal{M} \subset \mathbb{R}^n$ is an embedded submanifold of Euclidean space equipped with Euclidean metric, $X$ and $Y$ can be extended in $\mathbb{R}^n$. Applying ambient covariant derivative $\tilde{\nabla}$ and decomposing it into tangential and normal components gives $\tilde{\nabla}_X Y = (\tilde{\nabla}_X Y)^T + (\tilde{\nabla}_X Y)^\perp$. The *second fundamental form* [43] is $\mathrm{II} : \mathfrak{X}(\mathcal{M}) \times \mathfrak{X}(\mathcal{M}) \to \Gamma(N\mathcal{M})$ given by $\mathrm{II}(X, Y) = (\tilde{\nabla}_X Y)^\perp$ where $\Gamma(N\mathcal{M})$ denotes the space of smooth normal vector fields.

## 2.3 Technical Assumptions

We now introduce a series of assumptions about the problem setting, all of which are considered standard in the relevant literature and are needed for our analytical results. We have the following assumption on the manifold and its second fundamental form.

**Assumption 2.4.** We assume that the manifold $\mathcal{M}$ is an embedded submanifold of the Euclidean space and that the norm of the second fundamental form is bounded by $C$ for all unit vectors $\xi, \eta \in T\mathcal{M}$, i.e., for all $\|\xi\| = \|\eta\| = 1$ we have $\|\mathrm{II}(\xi, \eta)\| \leq C$. This implies that the curvature

tensor $R(X,Y)Z$ is bounded by a constant $K_c$, i.e., $\|R(X,Y)Z\| \leq K_c\|X\|\|Y\|\|Z\|$, and the sectional curvature is bounded by a constant $K_s$ for any vector fields $X, Y, Z \in \mathfrak{X}(\mathcal{M})$.

The above assumption on the bounded second fundamental form connects the extrinsic and intrinsic geometries. As discussed in [42], it is a stronger condition than bounded sectional curvature, but it is required to measure the extrinsic difference on vectors, caused by transport operations. For more discussion, see Appendix C. In Riemannian optimization, variable updates may involve exponential mappings and retractions. Although exponential mapping has theoretically favorable properties, retractions are preferred in practice due to their computational efficiency. We have the following assumptions on retraction curves.

**Assumption 2.5.** For the retraction curve $\mathrm{Retr}_x(t\xi)$ with $\xi \in T_x\mathcal{M}$, we assume that

$$\left\|\frac{d}{dt}\mathrm{Retr}_x(t\xi)\right\| \leq \|\xi\| \quad \text{and} \quad \left\|\frac{d^2}{dt^2}\mathrm{Retr}_x(t\xi)\right\| \leq C'\|\xi\|^2,$$

where $C'$ is a constant depending on the manifold properties.

Since retractions are approximations of exponential mappings, the above assumption can be thought as the cost of applying retractions instead of exponential mappings. It is presented in a simplified form for analytical clarity; however, the proposed implementation can be readily extended to more general retraction-based settings, such as projection-based or smooth first-order retractions (see Appendix B). Assumption 2.5 is satisfied by various matrix manifolds and commonly used retraction curves. In particular, for exponential mappings on any manifold $\mathcal{M}$, the first condition holds with equality, while the second condition holds with $C' = C$, where $C$ is defined in Assumption 2.4. On the Stiefel manifold $St(p,n) = \{X \in \mathbb{R}^{n\times p} : X^\top X = I_p\}$, for polar decomposition retraction $\mathrm{Retr}_X(t\xi) = (X + t\xi)(I_p + t^2\xi^\top\xi)^{-\frac{1}{2}}$ Assumption 2.5 holds with $C' = 1$ (see Appendix B).

Next, we provide the following standard assumptions on the objective function, applicable to a diverse range of commonly used objectives.

**Assumption 2.6.** We assume that the objective function has the form $f(x) = \mathbb{E}_\nu[F(x,\nu)]$, where $\nu$ denotes the random index. The stochastic component of the function $F(\cdot,\nu) : \mathcal{M} \to \mathbb{R}$ is $L(\nu)$-Lipschitz for any $\nu$, i.e., it holds that

$$|F(x,\nu) - F(y,\nu)| \leq L(\nu)\mathrm{dist}(x,y),$$

for any $x, y \in \mathcal{M}$. $L(\nu)$ has a bounded second moment such that $\mathbb{E}_\nu[L(\nu)^2] \leq L^2$. We also assume that $f$ is lower bounded on $\mathcal{M}$, i.e., $\inf_{x\in\mathcal{M}} f(x) > -\infty$.

**Assumption 2.7.** We assume that the stochastic Riemannian oracle returns unbiased estimates, i.e.,

$$\mathbb{E}_\nu[\mathrm{grad}F(x,\nu)] = \mathrm{grad}f(x).$$

Furthermore, we assume that the second moment of the stochastic gradient is bounded such that $\mathbb{E}_\nu\left[\left\|\mathrm{grad}F(x,\nu)\right\|^2\right] \leq G^2$, and we denote its variance by $\sigma^2$.

Assumption 2.7 is standard in stochastic optimization with first order oracles [24]. With the definitions and assumptions provided above, we introduce the following lemma, which extends the result of [43, Theorem 4.1] on geodesic paths to broken geodesic paths.

**Lemma 2.8.** *Suppose $\mathcal{M}$ is an embedded submanifold of the Euclidean space with a second fundamental form bounded by $C$. Let $\gamma : [0,t] \to \mathcal{M}$ be a broken geodesic (a piecewise smooth curve with a finite number of curve segments, each of which is a geodesic) and $v \in T_{\gamma(0)}\mathcal{M}$. Then, we have*

$$\left\|\mathcal{P}_{0,t}^\gamma(v) - \mathrm{Proj}_{T_{\gamma(t)}\mathcal{M}}(v)\right\| \leq C\|v\|\mathrm{length}(\gamma),$$

*where the parallel transport is computed along the path $\gamma$.*

This result allows us to measure the distortion between an intrinsic parallel transport operation and an extrinsic projection operation on the manifold. We note that for the Euclidean case, the second fundamental form is 0. For the extension of Euclidean algorithms to the Riemannian setting, Lemma 2.8 characterizes the extra terms raised by the Riemannian geometry.

For a sequence of points $S_t = \{x_s\}_{s=0}^t$, let us define the parallel transport operator over $S_t$ as $\mathcal{P}_{S_t}^s := P_{x_{t-1},x_t}^g \circ P_{x_{t-2},x_{t-1}}^g \circ \dots \circ P_{x_0,x_1}^g$ and its inverse operator as $(\mathcal{P}_{S_t}^s)^{-1}$. In general, $\left\|\mathrm{grad}f(y)\right\|_\delta$ is difficult to compute directly. The following lemma provides an analyzable upper bound on $\left\|\mathrm{grad}f(y)\right\|_\delta$, which will be used in our analysis.

**Lemma 2.9.** *Let Assumption 2.4 hold. For a sequence of points $\{x_t\}_{t=0}^{T-1}$ which satisfies* $\text{dist}(x_t, x_{t+1}) \leq D \quad \forall t \in \{0, 1, \ldots, T-1\}$, *a set of gradient vectors* $\nabla_t := \text{grad} f(w_t) \in T_{w_t}\mathcal{M}$ *with* $\|\nabla_t\| \leq L$, $\text{dist}(x_{t+1}, w_t) \leq D$, *and a point $y$ that satisfies* $\text{dist}(w_t, y) \leq \delta := DT, \quad \forall t \in \{0, 1, \ldots, T-1\}$, *we have that*

$$\left\|\text{grad} f(y)\right\|_\delta \leq \left\|\frac{1}{T} \sum_{t=0}^{T-1} (\mathcal{P}_{S_{t+1}}^s)^{-1} \circ P_{w_t, x_{t+1}}^g (\nabla_t)\right\| + 3L\delta C.$$

The summands all live in $T_{x_0}\mathcal{M}$, so they can be summed. In the following sections, we design our algorithms and provide analytical bounds for the right-hand side of Lemma 2.9.

# 3 First Order Setting

In this section, we develop algorithms to find $(\delta, \epsilon)$-stationary points of nonsmooth nonconvex stochastic objectives and provide theoretical guarantees for their finite-time convergence rates. Our algorithms adapt the O2NC conversion of [13] to the Riemannian setting, and instead of directly implementing online Riemannian optimization, our specific formulation of the problem allows us to utilize techniques of Euclidean online algorithms with transport operations between tangent spaces.

## 3.1 The Structure of RO2NC versus O2NC

To understand O2NC, let us observe that for any algorithm with the update rule $x_{t+1} = x_t + \Delta_t$, one can write $f(x_{t+1}) = f(x_t) + \langle \Delta_t, \nabla_t \rangle$ where $\nabla_t = \int_0^1 \nabla f(x_t + s\Delta_t) ds$. For Euclidean problems, the O2NC approach proposed by [13] designed an algorithm to overcome the nonconvexity by observing that

$$f(x_T) - f(x_0) = \sum_{t=0}^{T-1} \langle g_t, \Delta_t - u \rangle + \sum_{t=0}^{T-1} \langle \nabla_t - g_t, \Delta_t \rangle + \sum_{t=0}^{T-1} \langle g_t, u \rangle,$$

where $\Delta_t$ can be generated via an online learning algorithm and $g_t$ are the stochastic gradients provided to the online learning algorithm. The above equation holds for any $u$ in hindsight, so the first summation corresponds to the regret of the online algorithm. For the last term we have freedom to select a suitable $u$. Specifically, the selection of $u = -D \sum_{t=0}^{T-1} \mathbb{E}[g_t]/\|\sum_{t=0}^{T-1} \mathbb{E}[g_t]\|$ with a suitable parameter $D$ facilitates the Goldstein stationarity analysis. To have an efficient Riemannian adaptation for the O2NC algorithm, we encounter some major challenges:

(i) Since the iterates are constrained to be on the manifold, we must use retractions in the updates, so we have $x_{t+1} = \text{Retr}_{x_t}(\Delta_t)$. Also, for the gradient evaluation part, we use the update $w_t = \text{Retr}_{x_t}(s\Delta_t)$ for a randomly selected $s \sim \texttt{unif}[0, 1]$. In the Euclidean setting, two consecutive iterates are connected with a line segment as $x_{t+1} = x_t + \Delta_t$, parameterized to have zero acceleration. However, for manifolds with general retraction curves, we need to handle additional terms that contain the velocity and acceleration of the retraction curves.

(ii) Unlike the Euclidean case where we can directly add and subtract vectors, in the Riemannian setting different variables belong to various tangent spaces. Performing calculations using these variables necessitates operations such as parallel transport or projection, making the analysis more challenging compared to its Euclidean counterpart. This affects *both* algorithm design and technical analysis. For the design of RO2NC, we first consider the parallel transport in Section 3.2 and then consider a more efficient version of our algorithm with projections in Section 3.3. As for the analysis we cannot simply use the benchmark $u$ as in [13], and we must cleverly design an optimal $u$, elaborated in the next section.

## 3.2 RO2NC with Parallel Transport

In this section, we consider the RO2NC algorithm where the actions $\Delta_t$ are updated via parallel transport operations. The updates are similar in vein to online gradient descent and allow us to rework the Euclidean regret analysis with the use of parallel transports. The algorithm is formally stated in Algorithm 1 and works based on an inner loop indexed by $t$ (iteration) and an outer loop indexed by $k$ (epoch). In this section, except in the algorithm outline, we omit $k$ for notational convenience.

---
**Algorithm 1** Riemannian Online to NonConvex (RO2NC)
---
**Input:** $K \in \mathbb{N}$, $T \in \mathbb{N}$, initial point $x_{0,T} \in \mathcal{M}$, clipping parameter $D$, step size $\eta = D/G\sqrt{T}$
**for** $k = 1$ **to** $K$ **do**
    Initialize $x_{k,0} = x_{k-1,T}$, $\Delta_{k,0} = 0$
    **for** $t = 0$ **to** $T - 1$ **do**
        $x_{k,t+1} = \mathrm{Retr}_{x_{k,t}}(\Delta_{k,t})$
        $s_{k,t} \sim \mathtt{unif}[0, 1]$
        $w_{k,t} = \mathrm{Retr}_{x_{k,t}}(s_{k,t}\Delta_{k,t})$
        get gradient $g_{k,t} = \mathrm{grad}\, F(w_{k,t}, \nu_{k,t})$, where $\nu_{k,t}$ is a random index (based on data)
        **if** Using Parallel Transport **then**
            $g'_{k,t} = P^g_{w_{k,t}, x_{k,t+1}}(g_{k,t}) \in T_{x_{k,t+1}}\mathcal{M}$
            set $\Delta_{k,t+1} = P^g_{x_{k,t}, x_{k,t+1}}(\Delta_{k,t}) - \eta g'_{k,t}$
        **end if**
        **if** Using Projection **then**
            set $\Delta_{k,t+1} = \mathrm{Proj}_{T_{x_{k,t+1}}\mathcal{M}}(\Delta_{k,t} - \eta g_{k,t})$
        **end if**
        clip $\Delta_{k,t+1}$ on the convex set $\mathbb{B}_{T_{x_{k,t+1}}\mathcal{M}}(D)$
    **end for**
    Set $\bar{w}_k$ to $w_{k,\lfloor \frac{T}{2} \rfloor}$.
**end for**
Sample $w_{out} \sim \mathtt{unif}\{\bar{w}_1, ..., \bar{w}_K\}$
**Output:** $w_{out}$
---

At the $t$-th iteration in each epoch, we parallel transport both $\Delta_t$ and $g_t$ to update $\Delta_{t+1} = \mathrm{clip}(P^g_{x_t, x_{t+1}}(\Delta_t) - \eta g'_t)$, where $g'_t$ is the parallel transported version of $g_t$. We have the following convergence result for RO2NC.

**Theorem 3.1.** *Let $\delta, \epsilon \in (0, 1)$ and suppose that Assumptions 2.4,2.5,2.6,2.7 hold. Running Algorithm 1 with parallel transports for $N = KT$ rounds $T = O(\delta N)^{\frac{2}{3}}$ and $D = \delta/T$ gives an output that satisfies the following inequality*

$$\mathbb{E}[\|\mathrm{grad} f(w_{out})\|_\delta] \leq C_1(\delta N)^{-\frac{1}{3}} + 3\delta L C_2,$$

*where constants $C_1, C_2$ are given in the Appendix E.*

*Remark* 3.2. To find a $(\delta, \epsilon)$-stationary point, we can choose $\delta' = \min\{\delta, \frac{\epsilon}{6LC_2}\} \leq \delta$ to get a $(\delta', \epsilon)$-stationary point. Then, choosing $N = O(\delta^{-1}\epsilon^{-3})$ is sufficient for $\mathbb{E}[\|\mathrm{grad} f(w_{out})\|_\delta] \leq \epsilon$.

Theorem 3.1 indicates that for nonsmooth nonconvex optimization on Riemannian manifolds, RO2NC has the same complexity as its Euclidean counterpart [13]. The distortion caused by the curved geometry can be controlled with a suitable selection of parameters. Similar to the Euclidean setting, for smooth objectives, an $\epsilon$-stationary point can be found in $O(\epsilon^{-4})$ iterations by selecting $\delta = O(\epsilon)$. A key innovation in the analysis of Theorem 3.1 is the choice of

$$u_t = \mathcal{P}^s_{S_t}\left(-D\frac{\sum_{\tau=0}^{T-1}(\mathcal{P}^s_{S_{\tau+1}})^{-1} \circ P^g_{w_\tau, x_{\tau+1}}(\nabla_\tau)}{\|\sum_{\tau=0}^{T-1}(\mathcal{P}^s_{S_{\tau+1}})^{-1} \circ P^g_{w_\tau, x_{\tau+1}}(\nabla_\tau)\|}\right),$$

where $\nabla_t = \mathbb{E}[g_t]$. The main idea behind this selection is to transport the gradient vectors along the path $\{x_T, ..., x_1, x_0\}$ to create a base vector $u_0$ and then choose $u_t = \mathcal{P}^s_{S_t}(u_0)$. Although the best actions for Algorithm 1 are time-dependent, they can still be analyzed with parallel transports.

### 3.3 RO2NC with Projection

We now address the case where parallel transport operations are costly and RO2NC use projections instead. While more efficient, the introduction of projections bring forward technical challenges as they lack the isometry property (unlike parallel transports). In this case, at the $t$-th iteration in each epoch, we calculate $\Delta_t - \eta g_t$ in the ambient space and project it to the tangent space of $\mathcal{M}$ at iterate $x_{t+1}$ to obtain $\Delta_{t+1}$. The convergence of Algorithm 1 with projection operations is presented in the following theorem.

**Theorem 3.3.** *Let $\delta, \epsilon \in (0,1)$ and suppose that Assumptions 2.4,2.5,2.6,2.7 hold. Running Algorithm 1 with projections for $N = KT$ rounds with $T = O(\delta N)^{\frac{2}{3}}$ and $D = \delta/T$ gives an output that satisfies the following inequality*

$$\mathbb{E}[\|\mathrm{grad} f(w_{out})\|_\delta] \le C_3(\delta N)^{-\frac{1}{3}} + C_4 \delta^{\frac{1}{3}} N^{-\frac{2}{3}} + \delta LC,$$

*where constants $C_3, C_4$ are given in Appendix E.*

*Remark* 3.4. Theorem 3.3 implies that $N = O(\delta^{-1}\epsilon^{-3})$ and $\delta = O(\epsilon)$ is sufficient to have $\mathbb{E}[\|\mathrm{grad} f(w_{out})\|_\delta] \le \epsilon$, since $\delta < 1$ and $\delta^{\frac{1}{3}} N^{-\frac{2}{3}} < (\delta N)^{-\frac{1}{3}}$ order-wise. So, we can follow the same argument in Remark 3.2.

Theorem 3.3 shows that the same complexity can be achieved using projections instead of parallel transport operations, greatly improving the efficiency of RO2NC from an implementation perspective. While the implementation of the algorithm relies solely on projection operations, the analysis makes use of parallel transport to upper bound the term $\|\mathrm{grad} f(w_{out})\|_\delta$ in our theorem.

The projection-based analysis allows us to first calculate $u = -D \sum_{t=0}^{T-1} \nabla_t / \|\sum_{t=0}^{T-1} \nabla_t\|$ directly in the ambient space and then project it back to the tangent space to get $u_t = \mathrm{Proj}_{T_{x_t}\mathcal{M}}(u)$, which again highlights a key novelty in our analysis.

## 4 Zeroth Order Setting

In this section, we consider the case where gradient queries are unavailable and only noisy function evaluations can be obtained. In the context of online learning, this is analogous to a system with two-point bandit feedback. One common approach to address the nonsmooth problems in this setting is to derive a gradient estimator $g_\delta$ based on function values and use that in the gradient-based algorithms as a replacement of $\mathrm{grad} F$.

For zeroth order optimization in the Euclidean setting [37], the gradient estimator is constructed with $F(x \pm \delta u, \nu)$, where $u$ is uniformly sampled from a unit sphere. In the Riemannian setting, $x + \delta u$ is replaced with $\mathrm{Exp}_x(\delta u)$, and $u$ is sampled uniformly from a sphere in $T_x\mathcal{M}$. Then, the Riemannian gradient estimator is given as follows,

$$g_\delta(x) = \frac{d}{2\delta}(F(\mathrm{Exp}_x(\delta u), \nu) - F(\mathrm{Exp}_x(-\delta u), \nu))u, \tag{2}$$

where $d$ is the dimension of $T_x\mathcal{M}$. Two major problems arise from the Riemannian formulation: (i) In the Euclidean setting, we have $\nabla f_\delta(x) = \mathbb{E}_u[\nabla f(x + \delta u)] \in \partial_\delta f(x)$, which implies that the expectation of the gradient estimator is included in the Goldstein subdifferential set [45]. However, in the Riemannian setting, stating the relation between $\mathbb{E}_u[g_\delta(x)]$ and $\partial_\delta f(x)$ is a challenge in itself. (ii) Also, the geometric relation $\partial_v f_\rho(x) \subseteq \partial_{v+\rho} f(x)$ used in the Euclidean zeroth order analysis [37] does not hold in the Riemannian setting due to distortion caused by the manifold geometry.

We first define $h_\delta(x) := \int f \circ \mathrm{Exp}_x(u) dp_x(u)$, where $p_x$ is a uniform measure over $\mathbb{B}_{T_x\mathcal{M}}(\delta) \subset T_x\mathcal{M}$. By analyzing the relationship between $\mathrm{grad} h_\delta(x)$ and $\partial_\delta f(x)$ in Lemma A.5, we introduce the following lemma to address the above two challenges.

**Lemma 4.1.** *Consider a point $y$ and a set of points $\{x_t\}_{t=0}^{T-1}$ which satisfy $\mathrm{dist}(y, x_t) \le \frac{\delta}{2}$. The gradient estimator $g_\delta$ satisfies $\mathbb{E}_u[g_\delta(x_t)] \in \partial_\delta f(x_t) + \mathbb{B}_{T_x\mathcal{M}}(\frac{1}{3}K_s L \delta^2)$ and $P^g_{x_t, y}(\partial_{\frac{\delta}{2}} f(x_t)) \subset \partial_\delta f(y) + \mathbb{B}_{T_y\mathcal{M}}(2CL\delta)$, where $+$ denotes the Minkowski sum of two sets, $C$ denotes the bound on the second fundamental form (Assumption 2.4) and $K_s$ bounds the sectional curvature of the manifold.*

With the help of Lemmas 4.1 and A.5 we bound $\|\mathrm{grad} f(w_{out})\|_\delta$ in terms of $\|\mathrm{grad} h_{\frac{\delta}{2}}(w_{out})\|_{\frac{\delta}{2}}$, and we then employ that inequality for the following theorem, providing the finite-time convergence guarantee using the zeroth order gradient estimator.

**Theorem 4.2.** *Let $\delta, \epsilon \in (0,1)$ and suppose that Assumptions 2.4,2.5,2.6 hold. Running Algorithm 1 for $N = KT$ rounds with $T = O(\delta N)^{\frac{2}{3}}$ and $D = \delta/T$ using the zeroth order gradient estimator in equation 2 gives an output that satisfies*

$$\mathbb{E}[\|\mathrm{grad} f(w_{out})\|_\delta] \le C_5(\delta N)^{-\frac{1}{3}} + \delta LC_6,$$

*where $C_5$ and $C_6$ are given in Appendix E.*

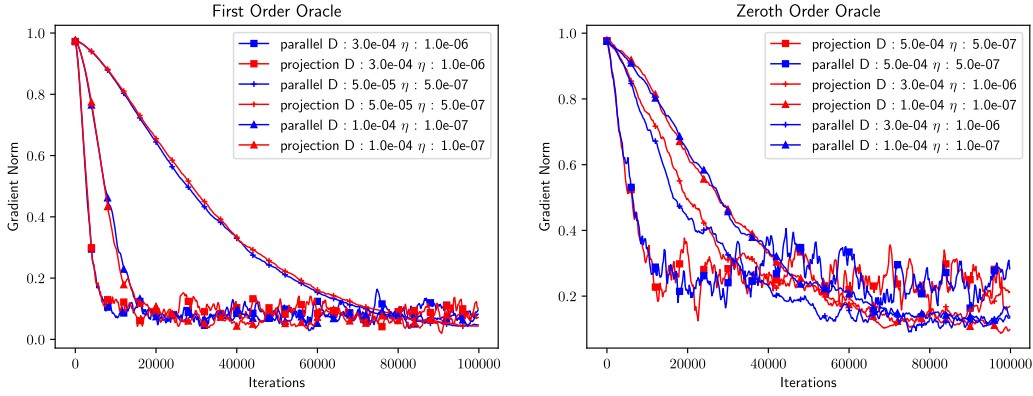

Figure 1: Evaluation of gradient norms in both settings; $D$: clipping parameter, $\eta$: step size.

The theorem considers Algorithm 1 with parallel transports, but a similar result can be obtained with projections. In our analysis, we show that although extra terms arise due to the introduction of zero order gradient estimator and the approximation to Goldstein subdifferential sets, these additional terms can be controlled by a careful adjustment of $\delta$. Consequently, the overall iteration complexity with respect to $(\delta,\epsilon)$ can be maintained in the zeroth order setting.

## 5 Numerical Experiments

We provide the following numerical experiments to validate our results.

**Model and Setup.** We consider the sparse principal component problem defined on the Euclidean unit sphere $\mathbb{S}^{n-1}$ in $\mathbb{R}^n$. The parallel transport operations have closed-form solutions on spheres. The objective function can be written as $\min_{x\in\mathbb{S}^{n-1}}\{-x^\top A x + \mu\|x\|_1\}$, where $A = \mathbb{E}[\nu\nu^\top]$ and $\nu \sim \mathcal{N}(0, A)$ is sampled from a multivariate Gaussian distribution. For RO2NC, we choose our retraction curves to be $\mathrm{Retr}_x(v) = (x + v)/\|x + v\|$, and we use exponential mappings for calculating the gradient estimator in ZO-RO2NC.

**Evaluation and Results.** Since the direct evaluation of the Goldstein subdifferential set and $\|\mathrm{grad}f(w_{out})\|_\delta$ requires calculation over a convex hull, which is highly impractical, we instead evaluate $\left\|\frac{1}{T}\sum_{t=0}^{T-1}(\mathcal{P}_{S_{t+1}}^s)^{-1}\circ P_{w_t,x_{t+1}}^g(\mathrm{grad}f(w_t))\right\|$ as an upper bound.

In our experiments for RO2NC, we illustrate the decay of the gradient norms. We run Algorithm 1 using both parallel transport and projection operations for $K = 500$ epochs, each consisting of $T = 200$ iterations. Convergence of the algorithms depends on the selection of parameters $D$ and $\eta$. For both settings, $\eta$ is chosen orders of magnitude smaller than $D$, and the plot is reported in Fig. 1. We can see that the performance of the projection approach is comparable with that of parallel transport approach. Optimization with the zeroth order oracle is more sensitive to the selection of parameters and the convergence is slower than the first order setting, but with a suitable choice of parameters ZO-RO2NC indeed converges.

## 6 Conclusion, Limitations, and Future Work

We addressed the finite-time analysis of nonsmooth, nonconvex stochastic Riemannian optimization. We proposed the RO2NC algorithm, which is guaranteed to find the Riemannian extension of $(\delta, \epsilon)$-Goldstein stationary points with $O(\delta^{-1}\epsilon^{-3})$ sample complexity. When gradient information is unavailable, we introduced ZO-RO2NC with a zeroth order gradient estimator, which also achieves optimal sample complexity. While our stationarity condition is defined via parallel transport of the Clarke subdifferential, alternative notions, such as different transport maps or subdifferentials, may also be considered. Furthermore, a deeper analysis of curvature effects and their role in optimality remains an open direction. Finally, designing adaptive algorithms that can exploit curvature information more effectively can be an interesting problem for future research.

## Acknowledgments

The authors gratefully acknowledge the support of NSF ECCS-2240788 Award as well as Northeastern TIER 1 Program for this research.

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
