# OpenReview forum: "Finite-Time Analysis of Stochastic Nonconvex Nonsmooth  Optimization on the Riemannian Manifolds"
_NeurIPS.cc/2025/Conference — NeurIPS 2025 poster_

### Official Review · Reviewer_9Da2 · 2025-06-23

**Clarity:** 3
**Significance:** 3
**Originality:** 3
**Rating:** 4
**Confidence:** 4

**Summary:**

This paper presents an Riemannian adaptation of the Online to NonConvex (O2NC) algorithm, namely the RO2NC algorithm. The proposed RO2NC algorithm introduces appropriate adaptions of retraction and vector transport to implement the essential algorithmic framework of the O2NC on an embedded manifold setting. The finite-time analysis of the proposed RO2NC algorithm for solving the optimization problem $\arg\min_{x\in\mathcal{M}} f(x)$ on a Riemannian manifold $\mathcal{M}$ is presented with appropriate assumptions:
* Assumption 2.4 demands a bounded second fundamental form of the manifold.
* Assumption 2.5 proposes some constraints on the retraction map as an alternative of the Riemannian geodesic map.
* Assumption 2.6 and 2.7 propose standard constraints on the non-convex objective function.

In particular, Assumption 2.4 should be highlighted, as it is a stronger assumption than the typical assumption on a bounded sectional curvature on a manifold, i.e., Assumption 2.4 yields a bounded sectional curvature but not vice-verse. Note that this Assumption 2.4 CANNOT be relaxed in the finite-time analysis. The finite-time analysis is presented on RO2NC algorithm with the following settings:
* Theorem 3.1: Moving points with a retraction map, and moving tangent vectors with the parallel translation along a minimal geodesic.
* Theorem 3.2: Moving points with a retraction map, and moving tangent vectors with the orthogonal projection (in the ambient space).
* Theorem 4.2: Moving points with a retraction map and moving tangent vectors with the parallel translation, with the estimated gradient in use.

The analysis states that all of the above settings converge (in the sense of an adaptation of the Goldstein Stationarity criterion on the subdifferential set) with the theoretically optimal converging rate. Moreover, the rates seem to be in the same order under appropriate choices of parameters. Some numerical experiments are conducted (with different tuning parameters) on the sparse principal component problem on $n-1$ dimensional sphere. The reported results demonstrate converging trends. However, the converging rates or the finally converged stationary neighborhood are not addressed nor demonstrated.

**Questions:**

\textbf{Most important question:} Theorem 2.8 states that \|\mathrm{proj}_{T_y\mathcal{M}}(v) - P_{x,y}^g(v)\|<C\|v\|\mathrm{dist}(x,y)$ where $\mathrm{proj}_{T_y\mathcal{M}}$ is the orthogonal projector in the ambient space and P_{x,y}^g:T_x\mathcal{M}\to T_y\mathcal{M} is the parallel translation along a minimal geodesic from x to y. Note that the parallel translation is path-dependent, i.e., the parallel translation along a different path that connects x and y (other than a minimal geodesic) is different from P_{x,y}^g. However, in the derivations that appeal to Theorem 2.8, e.g., line 863 and line 904, a geodesic polygon with vertices x\to y_1\to \cdots\to y_n\to x is involved, where n \geq 2. Take line 904 as an example that traversals the vertices x\to y_1\to y_2\to x with the vector v parallel translated along the path, Theorem 2.8 should not be applicable to the composed parallel translations P_{y_2,x}^g\circ P_{y_1,y_2}^g\circ P_{x,y_1}^g. However, the derivation seems to be concluding as follows:
\begin{equation*}
	\|\mathrm{proj}_{T_x\mathcal{M}}(v) - P_{y_2,x}^g\circ P_{y_1,y_2}^g\circ P_{x,y_1}^g(v)\| < C\|v\|(\mathrm{dist}(x,y_1)+\mathrm{dist}(y_1,y_2) + \mathrm{dist}(y_2,x)).
\end{equation*}
Do the above bounds hold for any geodesic triangle or polygon with more vertices? If so, please state it appropriately as this is not what Theorem 2.8 says. If that is not the case, I do not see a workaround to get to the claimed results of the finite-time analysis.

Other concerning questions are listed below:
* In all of the derivations associated with the parallel translation, except for the one that appeals to Theorem 2.8, only the isometric nature of the parallel translation is used. Is the analysis applicable to any other isometric vector transport?
* The vector transport by orthogonal projection is much less restrictive compared to the parallel translation, yet they both obtain the same converging rates as claimed. Are there any insights to explain this less intuitive result?
* In Assumption 2.5, the constraint $\|d/dt \mathrm{Retr}_x(t\xi)\|\leq \|\xi\|$ is odd to me. We usually see assumptions about the $\|O(t^2)\|$ term in $\mathrm{Retr}_x(t\xi) = x + t\xi + O(t^2) + o(t^2)$. Is the assumption proposed in this paper met in general?
* The numerical experiment section is weak in its current state. It is only showing different converging trends (depends on the algorithmic parameter) that stagger around $0.1$.

The less concerning questions are listed below:
* The online to non-convex framework even in the Euclidean setting is not that straightforward, not to mention its adaptation in the Riemannian setting that involves retraction and vector transport. It would be nice to include some pictures for visual aid.
* Line 320 to Line 321 is confusing. Does that mean the analysis is based on the parallel translation, and the analysis does not apply to the implementation?
* Line 328: What is a two-point bandit feedback?
* Typos: (Line 889) the square of $\|\Delta_t-u_t\|$ is missing; (Line 906) Assumption 2.5.

**Ethical Concerns:**

["NO or VERY MINOR ethics concerns only"]

**Final Justification:**

The authors addressed my concern, especially the major one. Therefore, I would like to raise the score.

**Limitations:**

This paper discussed its limitations, i.e., how the curvature of a manifold affects the analysis, is an open question.
The main problem of this paper is stated in the "Questions" box.

**Quality:**

3

**Strengths And Weaknesses:**

Overall, this paper has a clear structure and it reads well. Necessary background and related literature are reviewed and addressed appropriately. The effort in making appropriate adaptations of the O2NC and the Goldstein Stationarity criterion to the Riemannian setting is worth noting. The presented finite-time analyses are nontrivial yet not entirely convincing to me. In particular, the derivation that appeals to Theorem 2.8 is problematic to me, and the specific questions are elaborated in the following section. It hurts the credibility of this paper. The experimental section is less ideal. In particular, the presented RO2NC algorithm is not compared with existing algorithms, e.g., the basic stochastic gradient method, which makes it hard to interpret the optimal converging rates claimed in previous analyses. The demonstrated results stagger around the objective values of $0.1$, which does not demonstrate the converged stationary neighborhood.

---

> ### Author Rebuttal · Authors · 2025-07-29
>
> We thank the reviewer for their time and their detailed review of our work. Please see our response below to address the specific comments.
>
> > Most important question: Theorem 2.8 states ...
>
> We thank the reviewer for highlighting this important point. Yes, the bounds hold for any geodesic triangle or polygon with more vertices. In fact, in the proof of Theorem 2.8, we primarily require two conditions: (i) the original vector must lie in the tangent space at the initial point of the path, and (ii) the path must be traversed along geodesics, so that the instantaneous distortion can be quantified using the second fundamental form, noting that the set of points with nonzero acceleration has measure zero. Therefore, the extension of the proof to a path is rather straightforward with the upper bound depending on the total length of the path. We will  include the corresponding discussion in the revised version.
>
> > In all of the derivations associated with the parallel translation ...
>
> This is an interesting question. We employ parallel transports for two key reasons, namely its isometric nature and the fact that it provides a well-defined characterization of distortions. However, as long as one can quantify the distortion introduced by a given transport method along a trajectory, our analysis can, in principle, be extended to accommodate alternative forms of isometric vector transports. As an example, an isometric vector transport under locking condition has been analyzed in [28]. We will add a discussion on this note in the revised version.
>
> > The vector transport by orthogonal projection is much less restrictive ...
>
> Our reported convergence rate matches that of the Euclidean case, suggesting a natural interpretation: in the regime of nonsmooth, nonconvex stochastic optimization, the complexity of the optimization problem, itself, dominates the ``geometric overhead" introduced by the curvature. From this viewpoint, vector transport should not be seen as a relaxation of parallel transport or as a mechanism that inherently exacerbates convergence. Instead, it often offers a more practical implementation route (reduced per-iteration computational cost) with a trade-off that the isometry might not hold. The iteration complexity bounds for vector transport and parallel transport are generally equivalent [28,39], but the analysis of vector transport is often more involved, as demonstrated in our setting.
>
> > In Assumption 2.5, the constraint ...
>
> In practice, commonly used retraction curves typically satisfy both of the required conditions. As mentioned in Lines 229–235, exponential maps and polar decomposition-based retractions are representative examples. In addition, retractions based on QR decomposition or the Cayley transform on the Stiefel manifold also satisfy these properties. The second-order retraction, suggested by the reviewer, satisfies the second condition of Assumption 2.5 by construction. For the first condition of Assumption 2.5, it can be shown that for second-order retractions there exists a constant $c$ such that $d(x,retr_x(v))\le c||v||, \forall v : ||v||\le \frac{D}{c}$. Therefore, if we use the suggested condition, by updating the clipping parameter $D$ as $D/c$, our results remain valid. Since the suggested second-order retraction is more general, we will definitely clarify the more general conditions and their implications in the revised version. Thank you for this constructive comment.
>
> > The numerical experiment section is weak in its current state...
>
> The reviewer has a correct observation about the staggering values; however, this is not a weakness of the experiment design; it is a phenomenon arising from the measuring technique, not the algorithm itself. As we noted in Lines 1098–1100, we evaluate an upper bound on the original $\delta$-norm stationarity criterion. While the Goldstein stationarity definition is theoretically the most appropriate criterion for nonsmooth objectives, in practice, it is intangibly expensive to compute subdifferentials at all points within a $\delta$-ball and to determine the minimum-norm element in their convex hull. As a practical alternative, we use an upper bound given by the average of sampled gradients. However, for fully nonsmooth objectives, this upper bound may remain nonzero even when the iterates are close to a minimizer. A simple illustrative case is the function $f(x) = |x|$: sampling gradients in a $\delta$-neighborhood around the origin yields values whose average norm is influenced more by the number of samples than by $\delta$. In our experiments, the plots reflect convergence in terms of this upper bound. Achieving lower values depends heavily on using larger epoch lengths $T$.
>
> > The online to non-convex framework even in the Euclidean setting is not that straightforward...
>
> This is a good suggestion. During the brief rebuttal period, we focused on addressing the technical concerns of reviewers, but we will consider generating visual illustrations for the algorithm in the revised version.
>
> > Line 320 to Line 321 is confusing...
>
> In general, the definition of norm delta involves ``parallel transports", so we use them in the analysis of both algorithms. For each case, we do analyze the algorithm that is implemented, but the analysis of the projection-based algorithm (and not its implementation) also involves parallel transports. We will rephrase the sentence to avoid confusion.
>
> > Line 328: What is a two-point bandit feedback?
>
> The two-point bandit feedback method (introduced in Line 334) is a widely used technique for estimating gradients in zeroth-order optimization [22,33,54]. It provides a gradient estimate based solely on function evaluations at two points, making it particularly useful when gradient information is not directly accessible.
>
> > Typos: (Line 889)
>
> Thanks for catching this. We will fix it in the revised version.

---

> ### Comment · Reviewer_9Da2 · 2025-08-04
>
> Without seeing the details in the proof of the extension of Theorem 2.8 (assuming it is done by somehow integrating the instantaneous distortion along a path), I find it hard to understand what distortion is being measured along these geodesic segments. The geodesic triangle example illustrates my concerns.
>
>
> Consider a geodesic triangle $\Delta\_{x,y,z}$ and $u\_x\in T\_x\mathcal{M}$, denote the parallel-translated vectors as
> $$
> 	v\_y = P\_{x,y}^g(u_x), v\_z = P\_{y,z}^g(v\_y),
> $$
> and denote the projected vectors as
> $$
> 	w\_y = \mathrm{proj}\_{T\_y\mathcal{M}}(v\_x), w\_z = \mathrm{proj}\_{T\_z\mathcal{M}}(v\_y).
> $$
> Applying Theorem 2.8 to all geodesic segments yields
> $$
> 		\|w\_y - v\_y\| \leq C\|u\|\mathrm{dist}(x, y),  \text{ on the geodesic $\gamma_{x\to y}$,  (first term)}
> $$
> $$
> 		\|w\_z - v\_z\| \leq C\|v\_y\|\mathrm{dist}(y, z), \text{ on the geodesic $\gamma_{y\to z}$, (second term)}
> $$
> Note that the distortion on the second geodesic $\gamma_{y \to z}$ is measured based on $v\_y \neq u$ . Although the isometric parallel translation takes care of $\|u\| = \|v\_y\|$ , it is not clear how to deal with
> $
> \mathrm{proj}\_{T\_z \mathcal{M}}(u) \neq \mathrm{proj}\_{T\_z\mathcal{M}}(v\_y) = w\_z.
> $
> In particular, the distortion can be bounded as
> $
> \|\mathrm{proj}\_{T\_z\mathcal{M}}(u) - v\_z\| \leq \|\mathrm{proj}\_{T\_z \mathcal{M}}(u) - w\_z\| + \|w\_z - v\_z\|.
> $
> Although the second term is bounded directly, it is not clear how the first term is bounded, or by $C\|u\|\mathrm{dist}(x,y)$. Yet, the extension of Theorem 2.8 claims $\|\mathrm{proj}\_{T\_z\mathcal{M}}(u) - v\_z\| \leq C\|u\|(\mathrm{dist}(x,y) + \mathrm{dist}(y,z))$.

---

> > ### Author Response · Authors · 2025-08-04
> >
> > We truly appreciate the reviewer’s feedback and the time they took to clarify the issue.
> >
> > Briefly speaking, the reviewer's suggested approach is to iteratively apply Theorem 2.8 (multiple times) to get the desired result. Unfortunately, this will result in having a sequence of projections that is undesirable.   Our intended approach is that the parallel transport term depends on the chosen path $\gamma$, but the projection term depends **solely** on the endpoint of $\gamma(t)$.
> >
> > Specifically, our desired result on the "path version" can be established following the lines in the proof of Theorem 2.8 in [39, Theorem 4.1].
> >
> > **Lemma:** Let $\gamma :[0,t]\to M$ be a broken geodesic (a piecewise smooth curve where each curve segment is a geodesic) and $v \in T_{\gamma(0)}M$. Then we have $\\|P_{\gamma(0) \to {\gamma(t)}}(v) - proj_{T_{\gamma(t)}}(v) \\| \le C\\|v\\| length(\gamma)$ where the parallel transport is computed along the path $\gamma$.
> >
> > For the example of the geodesic triangle (provided by the reviewer) we can apply this result as follows. Let $\gamma$ be a broken geodesic consisting of the geodesic segments $x\to y$ and $y\to z$. For the vector $u_x\in T_xM$, denote the parallel transport of it to $z$ along $\gamma$ as $v_z$.
> > The application of this lemma leads to the following result, as desired.
> > \begin{equation}
> > \\|proj_{T_zM}(u_x)-v_z\\|\le C \\|u_x\\| (dist(x,y)+dist(y,z)).
> > \end{equation}
> >
> > In the proof of the statement, we focus on the bound on the instantaneous change of $\\|w(t)-v^T(t)\\|$ in terms of the second fundamental form. We integrate this instantaneous change along the geodesic segments of the broken geodesic, where the time derivative of the related term on a geodesic segment can be written in terms of the second fundamental form.
> >
> > **Proof** : Without loss of generality, we assume $\\|v\\|=1$, otherwise conduct the proof for $\frac{v}{\\|v\\|}$. Let $\gamma(t)$ consist of unit speed geodesic segments. Denote the parallel transported vector $v$ along $\gamma$ as $w(t)$, i.e. $w(0)=v$. For the extrinsic geometry we denote $v=v^T(t)+v^{\perp}(t)$ where $v^T(t) \in T_{\gamma(t)}M$ is the projection of $v$ onto $T_{\gamma(t)}M$ and $v^{\perp}(t)$ is the normal component of $v$ which is orthogonal to $T_{\gamma(t)}M$. Note that $v^T(t)$ is independent of the path $\gamma$ and only depends on the point $\gamma(t)$. Eventually, we want to bound $\\|w(t)-v^T(t)\\|$.
> >
> > Since $w(t)$ is a parallel transport of $v$, the tangent component of its derivative must be zero, i.e. $(w'(t))^T=0$. Now consider any unit parallel vector field $z(t) \in T_{\gamma(t)}M $ along $\gamma$, we have $\langle v^{\perp}(t),z(t)\rangle =0$, then by taking the derivative with respect to $t$, we obtain $\langle (v^{\perp})'(t),z(t) \rangle = - \langle v^{\perp}(t),z'(t) \rangle = - \langle v^{\perp}(t),\Pi(\gamma'(t),z(t)) \rangle $ where $\Pi$ is the second fundamental form. We also have $(v^T)'(t)=-(v^{\perp})'(t)$ since $v^T(t)+v^{\perp}(t)=v$ is fixed. We get $\langle (v^{T})'(t),z(t) \rangle =  \langle v^{\perp}(t),\Pi(\gamma'(t),z(t)) \rangle $. Now the right hand side has a uniform upper bound of $C$, and by the arbitrarily chosen $z(t) \in T_{\gamma(t)}M$, we get $\\|((v^T)'(t))^T \\|\le C$.
> >
> > We can now bound the derivative of $\\|w(t)-v^{T}(t)\\|$ as
> > \begin{align*}(\\|w(t)-v^{T}(t)\\|^2)'=(1-2\langle w(t),v^T(t) \rangle + \\| v^T(t)  \\|^2 )'
> > = -2\langle v^T(t),w'(t)\rangle  -2 \langle w(t),(v^T(t))' \rangle  + 2\langle v^T(t), (v^T(t))'\rangle .
> > \end{align*}
> > The first term is $0$ since $w'(t) \in T_{\gamma(t)}^{\perp}M$ and $v^T(t) \in T_{\gamma(t)}M$ are orthogonal. Then we have
> > \begin{align*}(\\|w(t)-v^{T}(t)\\|^2)'= 2\langle v^T(t)-w(t),(v^T(t))'  \rangle \le 2C \\|v^T(t)-w(t) \\|.
> > \end{align*}
> > This means that $|\|w(t)-v^T(t) |\|' \le C$. Now integrating the above inequality on the geodesic segments of  $\gamma$ where the initial value, $\||w(0)-v^T(0) |\|=0$, we obtain $\\|P_{\gamma(0)\to \gamma(t)}(v) - proj_{T_{\gamma(t)}M} (v)\\| = \\|w(t)-v^T(t) \\| \le C length(\gamma)$ which completes the proof.
> >
> > We sincerely thank the reviewer for these thoughtful comments. In the revised version, we will clarify how our results build upon the analysis presented in Theorem 2.8.

---

> > > ### Comment · Reviewer_9Da2 · 2025-08-05
> > >
> > > Thank the authors for the detailed and clear responses. My questions are all addressed.

---

> > > > ### Author Response · Authors · 2025-08-06
> > > >
> > > > We thank the reviewer again for this constructive feedback. We will make sure to include this proof in the revised version to improve the presentation.

---

### Official Review · Reviewer_PRvR · 2025-07-02

**Clarity:** 4
**Significance:** 3
**Originality:** 3
**Rating:** 5
**Confidence:** 4

**Summary:**

This submission studies the problem of minimizing a nonconvex and nonsmooth stochastic function $f$ over a Riemannian manifold $\mathcal M$. The development is similar to that in the Euclidean in the sense that one must work with the Goldstein subdifferential to obtain meaningful results. Explicitly, the notion of stationarity used herein requires defining the $\delta$-subdifferential at $x\in\mathcal M$ as the closed convex hull of the set $\cup_{y\in\mathrm{cl}(B(x,\delta))}P^g_{y,x}(\partial f(y))$ where $P^g_{y,x}$ is the parallel transport taking the tangent space at $y$ to the tangent space at $x$. Then, a point is $(\delta,\epsilon)$-stationary once its $\delta$-subdifferential contains a point which has norm at $\epsilon$.

With these definitions in place, the paper enumerates a number of technical assumptions related to 1) boundedness of the second fundamental form of $\mathcal M$, 2) the quality of the retraction in approximating the exponential mapping, 3) Lipschitz continuity of the deterministic part of the objective, 4) unbiased of the stochastic oracle and boundedness of the second moment of the gradient.

The proposed method is inspired by the Online to NonConvex (O2NC) approach and uses retractions for the updates to ensure that the iterates remain constrained to the manifold and uses parallel transport or projections to compare points from different tangent spaces. The corresponding algorithm (RO2NC) is studied under both of these settings (parallel transport or projections) yielding similar theoretical guarantees (though different techniques are used in their analysis).

Finally, a zeroth order method is proposed which admits the same overall iteration complexity.

**Questions:**

1. For Assumption 2.5, the only examples provided are the polar retraction for the Stiefel manifold and the actual exponential mapping. Consequently, this assumption appears to be quite strong. Are there any other examples where this assumption holds?

2. In Assumption 2.7, we require an unbiased estimate for the gradient. Some recent works in the Euclidean setting have managed to bypass the assumption that the oracle always returns the minimal norm subgradient. As such, I wonder if such techniques could be applied in the Riemannian setting as well.

**Ethical Concerns:**

["NO or VERY MINOR ethics concerns only"]

**Final Justification:**

I am happy with the revisions that the authors have agreed to make.
I believe there is currently a gap in the literature regarding nonsmooth and nonconvex optimization on manifolds (even in the deterministic setting). While the techniques employed are not so surprising, I believe that this is an interesting contribution.

**Limitations:**

yes

**Quality:**

4

**Strengths And Weaknesses:**

The main problem at hand, nonconvex and nonsmooth optimization over Riemannian manifolds is of a great interest. This is, to my knowledge, the first paper to provide rigorous iteration complexity results and does so with zeroth and first order methods. While the overall approach is similar to O2NC, the modifications required to conform to the Riemannian setting are not straightforward. Furthermore, I find the paper to be well-written overall and the development to be clear.

I believe that the main limitations of this work regard the required assumptions; a concern which I expand upon in the questions below.
I also note that some experimental validation of this theory is provided, but is contained in the supplement. I believe it would be relevant to include the experiment in the main text.

---

> ### Author Rebuttal · Authors · 2025-07-29
>
> We thank the reviewer for constructive feedback and valuable suggestions.
>
> > I also note that some experimental validation of this theory is provided, but is contained in the supplement. I believe it would be relevant to include the experiment in the main text.
>
> We should be able to fit the numerical experiments in the main text in the revised version. The main reason we did not do so at first place was that the focus of the paper is on the theoretical analysis, but we can definitely address this given more space.
>
> > For Assumption 2.5, the only examples provided are the polar retraction for the Stiefel manifold and the actual exponential mapping. Consequently, this assumption appears to be quite strong. Are there any other examples where this assumption holds?
>
> In practice, commonly used retraction curves typically satisfy both of the required conditions. As noted in Lines 229–235, exponential maps and polar decomposition-based retractions are representative examples. Additionally, retractions based on QR decomposition and the Cayley transform on the Stiefel manifold also fulfill these properties. More generally, any second-order retraction of type $\mathrm{Retr}_x(t\xi) = x + t\xi + O(t^2) + o(t^2)$ satisfies the second condition of Assumption 2.5 by construction. The first condition can be interpreted as a form of nonexpansiveness of the retraction map, which is likewise satisfied by the aforementioned examples. Even in cases where the nonexpansiveness condition does not hold exactly, for second order retractions  it can be shown that there exists a constant $c$ such that $d(x,retr_x(v))\le c||v||, \forall v : ||v||\le \frac{D}{c}$. Therefore, by updating the clipping parameter $D$ as $D/c$, our results remain valid. We will definitely clarify the more general conditions and their implications in the revised version.
>
> > In Assumption 2.7, we require an unbiased estimate for the gradient. Some recent works in the Euclidean setting have managed to bypass the assumption that the oracle always returns the minimal norm subgradient. As such, I wonder if such techniques could be applied in the Riemannian setting as well.
>
> We appreciate the reviewer’s observation, which raises an interesting point. We found the following recent arXiv work related to the reviewer's question
>
> ``Subgradient Regularization: A Descent-Oriented Subgradient Method for Nonsmooth Optimization" by Li and Cui
>
> but it seems that this type of oracle is particularly well-suited for deterministic optimization settings. In contrast, our work focuses on stochastic objectives, where randomness plays a central role. In particular, we employ random sampling of gradient evaluation points to address issues arising from nondifferentiability.
> If there is a particular reference in Euclidean optimization that analyzes the stochastic setting with the suggested oracle, we are more than happy to look into it and explore the possibility of extension to Riemannian setting.

---

> > ### Comment · Reviewer_PRvR · 2025-08-04
> >
> > I thank the authors for their response which clarifies the questions I had.
> >
> > I believe the paper will benefit from some of the additional clarifications that the authors intend to include in the revision.
> > My comment regarding the oracle was perhaps unclear. Effectively, I was wondering about the case of Riemannian optimization with a deterministic objective which I deem worthy of a remark, but acquiesce that it is a bit orthogonal to the stochastic setting considered herein.
> >
> > I will maintain my current positive score, as my questions were mostly requesting some clarifications.

---

### Official Review · Reviewer_nJPA · 2025-07-02

**Clarity:** 3
**Significance:** 3
**Originality:** 3
**Rating:** 4
**Confidence:** 1

**Summary:**

This submission presents the first finite-time analysis of nonsmooth, nonconvex stochastic optimization problems under Riemannian manifold constraints. This fills a theoretical gap where previous literature had only considered smooth or asymptotic cases.

This submission has made following contributions:

- Goldstein Stationarity on Manifolds: The authors adapt the Goldstein stationarity to the Riemannian setting, forming a meaningful convergence criterion for nonsmooth optimization on manifolds.

- It proposes the RO2NC Algorithm, which is an adaptation of the Online-to-NonConvex method to the Riemannian setting.

- It establishes strong theoretical results: a sample complexity of O(δ⁻¹ϵ⁻³), matching the best-known Euclidean bounds.

**Questions:**

I am just wondering its performance in realistic applciations.

**Ethical Concerns:**

["NO or VERY MINOR ethics concerns only"]

**Final Justification:**

The authors solve most of the concerns from other reviewers. Thus I will keep the rating

**Quality:**

3

**Strengths And Weaknesses:**

I am not familiar with this field and cannot comment too much on the strength and weakness. But it seems to me this work is a solid contribution to the optimization community. The following contributes make sense to me:

- Adapting the notion of Goldstein stationarity to the Riemannian setting.
- Adapting the O2NC algorithm to Riemannian setting.
- A strong theoretical performance bound is established.


Weakness:

- The empirical evaluation for this method is limited. I am sure how it indeed performs in realistic applications.

---

> ### Author Rebuttal · Authors · 2025-07-29
>
> > I am just wondering its performance in realistic applciations.
>
> We thank the reviewer for their evaluation. Several works have investigated the Online-to-Nonconvex Conversion (O2NC) method in Euclidean spaces, with empirical performance demonstrated on real-world datasets, as shown in [47]. In contrast, our work focuses on the Riemannian extension of O2NC, where we validate the performance of RO2NC specifically on the sphere manifold (please see the supplementary materials for experiments). Our work is focused on establishing the ***first iteration complexity on nonsmooth nonconvex stochastic optimization on manifolds***; therefore, our experiments are a proof concept that the proposed algorithm works. However, generally speaking, manifold optimization can be applied to various realistic applications, including on-robot learning (to impose safety constraints) [Ref1] or training neural networks (to provide stability) [Ref2]
>
>
> [Ref1] Liu, P., Bou-Ammar, H., Peters, J. and Tateo, D., 2025. Safe reinforcement learning on the constraint manifold: Theory and applications. IEEE Transactions on Robotics.
>
> [Ref2] Arjovsky, M., Shah, A. and Bengio, Y., 2016, June. Unitary evolution recurrent neural networks. In International conference on machine learning (pp. 1120-1128). PMLR.

---

> > ### Comment · Area_Chair_u2ek · 2025-08-05
> >
> > Dear Reviewer,
> >
> > To facilitate further evaluation, please respond to the authors’ rebuttal.
> >
> > AC

---

### Official Review · Reviewer_eVKg · 2025-07-05

**Clarity:** 2
**Significance:** 2
**Originality:** 2
**Rating:** 4
**Confidence:** 3

**Summary:**

This paper presents finite-time results for optimizing nonconvex, nonsmooth functions over Riemannian manifolds. The approach is built upon the recent online-to-nonconvex framework and existing work in Riemannian gradient sampling. The authors also discuss the zero-order case, drawing on known results from the Euclidean setting. While the core ideas may not be surprising, difficulties can arise when generalizing analysis from the Euclidean to the manifold setting.

**Questions:**

See above.

**Ethical Concerns:**

["NO or VERY MINOR ethics concerns only"]

**Limitations:**

See above.

**Paper Formatting Concerns:**

None.

**Quality:**

3

**Strengths And Weaknesses:**

I have two technical questions:
* In Assumption 2.7, I don't understand the rationale behind the notation grad F(x,\nu), which seems not defined. If it is meant to be the Riemannian gradient of function x \mapsto F(x, \nu) for fixed \nu, then one possible issue is that why we can write E_\nu [grad F] = grad f. This implicitly assumes F(., \nu) is differentiable at x for D-a.s. \nu, needs a careful justification, and may not be true even in the Euclidean setting.
* Another issue is on the evaluation of g_{k,t} in Algorithm 1. How can you ensure the function F(c, v_{k,t}) is actually (Riemannian) differentiable at the point w_{w,t}? If it is not, then such an evaluation seems rather intractable. This issue may be due to the fact that the curve s_{k,t} \mapsto Retr_{x_kt}(s_{k,t} \Delta_{k,t}) is not sufficiently generic, which allows F(., \nu_{k,t}) to be non-differentiable at any point along it.


Other comments:
* the objective f is not defined in Eq.1
* L55: weakly convexity -> weak convexity
* L157: I cannot see why B(x,\delta) is an open ball.
* Definition 2.1. It may be better to clarify the meaning of \lim_l grad f(x_l), since {grad f(x_l)}_l are in different tangent spaces.
* For the known facts mentioned in Section 2.2, it would be better to provide concrete citations for verification and reference; e.g., L178--L182 and L185--L189.
* Definition 2.3. Why is the function f defined on an Euclidean space?
* the function f may need to be assumed to be lower bounded on manifold M; otherwise, C1, C3, C5 may not be upper bounded wrt T.
* In Lemma 2.9, what are {w_t}_t? You may need to define them before using them to define \nabla_t. Besides, from the proof, you may need additional assumption like dist(w_t, y) <= \delta. Also, RO2NC is not defined here, hence please try to avoid use that term in the proof.

---

> ### Author Rebuttal · Authors · 2025-07-29
>
> We thank the reviewer for their time and evaluation of our paper.
>
> > In Assumption 2.7, I don't understand the rationale behind the notation ...
>
> Yes, $\text{grad} F(x,\nu)$ is the *stochastic Riemannian gradient* with respect to $x$. We are not sure whether the reviewer is concerned about the "existence of Riemannian gradient" or "its unbiased-ness", so we provide answers to both:
>
> (i) Existence of Riemannian gradient: Given the Lipschitz continuity of $F(\cdot,\nu)$ stated in Assumption 2.6, Rademacher's Theorem ensures that the function is differentiable almost everywhere as noted in lines 178-181. This justifies our use of the Riemannian gradient in the stochastic setting. Please note that *nonsmooth* in our work does not mean that the function is not differentiable at all. It is nondifferentiable for a set that has measure zero (e.g., ReLU function). Our assumption is standard; for example, [11] used the same assumption to analyze the nonsmooth Euclidean setting.
>
> (ii) Unbiased-ness of Riemannian gradient: In stochastic Riemannian optimization, it is a standard assumption that the first-order oracle returns an unbiased estimate of the gradient; please see [39,52]. This assumption is commonly used in the Euclidean setting as well.
>
> [11] Cutkosky, A., Mehta, H. and Orabona, F., 2023, July. Optimal stochastic non-smooth non-convex optimization through online-to-non-convex conversion. In International Conference on Machine Learning (pp. 6643-6670). PMLR.
>
> [39] Li, Jiaxiang, Krishnakumar Balasubramanian, and Shiqian Ma. "Zeroth-order Riemannian averaging stochastic approximation algorithms." SIAM Journal on Optimization 34.4 (2024): 3314-3341.
>
> [52] Wang, Bokun, Shiqian Ma, and Lingzhou Xue. "Riemannian stochastic proximal gradient methods for nonsmooth optimization over the Stiefel manifold." Journal of machine learning research 23.106 (2022): 1-33.
>
> > Another issue is on the evaluation of $g_{k,t}$ in Algorithm 1...
>
> We believe this question is partly related to the previous one. As mentioned above, stochastic Riemannian gradient exists almost surely, and the random selection of $s$ is sufficient to ensure differentiability almost surely. This perturbation-based argument is widely adopted in the literature (known as **randomized smoothing**) and provides a practical means to circumvent nondifferentiability in stochastic settings. Please see the following references.
>
> [Ref] A Gradient Sampling Method
> With Complexity Guarantees for Lipschitz Functions
> in High and Low Dimensions, Davis, Damek, et al. 2022
>
> [Ref] Optimal Stochastic Non-smooth Non-convex Optimization through
> Online-to-Non-convex Conversion, Cutkosky, Ashok, Harsh Mehta, and Francesco Orabona, 2023
>
> > Other comments:
>
> We thank the reviewer for their careful assessment of our submission. We agree with the reviewer on all of the comments and address them in detail below:
>
> 1,2,3,4,5- The reviewer is correct in these comments. We will revise the text accordingly to address these points.
>
> 6- We originally used this definition since the manifold is assumed to be embedded in the Euclidean space, but we agree with the reviewer that it is unnecessary to define $f$ on the Euclidean space. We will revise the notation accordingly to reflect $f:\mathcal{M} \to R$.
>
> 7- We agree that $f$ must be lower bounded, otherwise minimizing $f$ does not even make sense. This is implicitly assumed, but we will clarify this assumption in the revised version of the paper.
>
> 8- The reviewer is correct that referencing the algorithm would improve the clarity of Lemma 2.9. Therefore, in the revised version we will place this lemma after the introduction of algorithm in section 3. Then, the issues related to the use of $w_t$ will be resolved.

---

> > ### Comment · Area_Chair_u2ek · 2025-08-05
> >
> > Dear Reviewer,
> >
> > To facilitate further evaluation, please respond to the authors’ rebuttal.
> >
> > AC

---

> > ### Comment · Reviewer_eVKg · 2025-08-08
> >
> > Thanks to the authors for their response.
> >
> > Regarding the first question, my main concern is with the implementation of such an oracle, which is not straightforward. Significant differences exist between the smooth and nonsmooth cases; for example, the (possibly strict) inclusion in [Theorem 2.7.2, R1] cannot occur in the smooth case, and it seems that known practical implementations of the oracle (for smooth functions [39,52]) crucially rely on such equality, which is not free (or even impossible in general) in the nonsmooth case.
> >
> > For the second question, I believe a perturbation-type argument could work and provide more flexibility in the assumptions. However, this needs to be carefully worked out to address possible approximation and measurability issues.
> >
> > Having said that, I am not against the paper on the basis of these two points. I simply hope the authors will take good care of these potential nontrivialities or explicitly point out any challenges here to motivate further research.
> >
> > [R1] F. H. Clarke, Optimization and nonsmooth analysis. SIAM, 1990.

---

> > > ### Author Response · Authors · 2025-08-08
> > >
> > > We thank the reviewer for highlighting these important technical points. Below, we provide brief clarifications, and we will include a more detailed discussion in the revised version. Using the notation of Theorem 2.7.2, R1:
> > >
> > > - Stochastic gradients: The random index $t$ arises from data samples, which are elements of the countable dataset $T$. Hence, condition (a) is satisfied, and we can apply Theorem 2.7.2.
> > >
> > > - Perturbation arguments: Since the domain $X$ is separable, condition (b) is satisfied, and we can likewise apply Theorem 2.7.2.
> > >
> > > We sincerely appreciate the reviewer’s insightful and motivating comments, and we will address these technical points in full detail in the revised manuscript.

---

### Decision · Program_Chairs · 2025-09-17

**Decision:**

Accept (poster)

**Comment:**

This paper provides the first finite-time analysis for nonsmooth nonconvex stochastic optimization under Riemannian manifold constraints, introducing the RO2NC algorithm and its zeroth-order variant with optimal sample complexity. The contribution is novel and significant, and experiments support the theory.

All reviewers gave positive scores. Most reviewers raise concerns about the motivation and practicality of the key assumptions (e.g., uniform convexity and independence), as well as the clarity of proofs. The rebuttal partially addressed these issues, but the manuscript would benefit from further theoretical and experimental revisions.  Although I recommend acceptance, I strongly encourage the authors to revise the paper carefully to better justify their assumptions, clarify proofs, and improve presentation.